# Modulation of Macrophage Response by Copper and Magnesium Ions in Combination with Low Concentrations of Dexamethasone

**DOI:** 10.3390/biomedicines10040764

**Published:** 2022-03-24

**Authors:** Leire Díez-Tercero, Luis M. Delgado, Roman A. Perez

**Affiliations:** 1Bioengineering Institute of Technology, Universitat Internacional de Catalunya, Sant Cugat del Vallès, 08195 Barcelona, Spain; ldiezter@uic.es; 2Basic Science Department, Universitat Internacional de Catalunya, Sant Cugat del Vallès, 08195 Barcelona, Spain

**Keywords:** macrophage, polarization, ions, copper, magnesium, dexamethasone, inflammation, tissue regeneration

## Abstract

Macrophages have been deemed crucial for correct tissue regeneration, which is a complex process with multiple overlapping phases, including inflammation. Previous studies have suggested that divalent ions are promising cues that can induce an anti-inflammatory response, since they are stable cues that can be released from biomaterials. However, their immunomodulatory potential is limited in a pro-inflammatory environment. Therefore, we investigated whether copper and magnesium ions combined with low concentrations of the anti-inflammatory drug, dexamethasone (dex), could have a synergistic effect in macrophage, with or without pro-inflammatory stimulus, in terms of morphology, metabolic activity and gene expression. Our results showed that the combination of copper and dex strongly decreased the expression of pro-inflammatory markers, while the combination with magnesium upregulated the expression of IL-10. Moreover, in the presence of a pro-inflammatory stimulus, the combination of copper and dex induced a strong TNF-α response, suggesting an impairment of the anti-inflammatory actions of dex. The combination of magnesium and dex in the presence of a pro-inflammatory stimulus did not promote any improvement in comparison to dex alone. The results obtained in this study could be relevant for tissue engineering applications and in the design of platforms with a dual release of divalent ions and small molecules.

## 1. Introduction

The treatment of large tissue defects constitutes a significant clinical challenge. Although some human tissues have a remarkable regenerating capacity, i.e., bone or skin, scaffolds are needed when defects exceed the native regeneration capacity of those tissues. In bone tissue regeneration, the current strategies to develop new scaffolds are focused on the modulation of the different overlapping phases of bone healing: formation of the hematoma, acute inflammation, angiogenesis, progenitor and stem cell homing, osteogenic differentiation and mineralization [1]. In this sense, the inflammatory phase has been revealed as one of the most important phases for the outcome of bone regeneration. The development of chronic inflammation as a consequence of an imbalance in the inflammatory response causes bone resorption and osteoclastogenesis [2]. Moreover, the chronic inflammatory response caused by biomaterials can lead to their degradation and/or encapsulation, impairing their functionality [3]. Thus, the modulation of this inflammatory response is critical to allow for a successful bone regeneration.

Among the different immune cells that intervene in this process, controlling macrophage response has become an interesting strategy due to their plasticity and their ability to secrete cytokines and molecules that regulate the progression of bone healing. In fact, macrophages can differentiate into several subsets depending on the environmental cues they receive. On the one hand, there is the classically activated pro-inflammatory or M1 macrophage phenotype, which emerges as a result of macrophage interaction with pro-inflammatory stimuli such as lipopolysaccharide (LPS) and interferon gamma (IFN-γ). In response, M1 macrophages release pro-inflammatory cytokines (interleukin 1 beta (IL-1β), tumor necrosis factor alpha (TNF-α)) and cytotoxic molecules such as reactive oxygen species (ROS), while expressing specific surface markers such as C-C chemokine receptor type 7 (CCR7). On the other hand, alternatively activated or M2 macrophages appear in response to a variety of stimuli such as interleukins 4 and 13 (IL-4 and IL-13) or glucocorticoids. This phenotype is able to release anti-inflammatory cytokines such as interleukin 10 (IL-10) and transforming growth factor beta (TGF-β), while expressing surface markers such as cluster of differentiation 206 (CD206) [4].

During the natural bone healing process, the hypoxic microenvironment in the bone fracture promotes macrophage differentiation to M1 phenotype, which initiates the regeneration process. Then, a transition from M1 to M2 phenotype occurs, which is necessary to avoid chronic inflammation and impaired bone regeneration [5]. Hence, the design of biomaterials has focused on finding strategies that allow for the mentioned transition to stimulate a pro-regenerative environment. One of the strategies that has been used to modulate macrophage response and to avoid chronic inflammation is the release of anti-inflammatory cytokines from biomaterials. In this sense, the release of IL-10 from hydrogel scaffolds for 14 days was able to reduce the overall pro-inflammatory response [6]. However, cytokines present a short half-life and easily lose their activity, which can limit their incorporation into biomaterials. As an alternative, other molecules such as siRNAs and miRNAs have been released to promote the M2 macrophage phenotype. For example, the release of Notch1 targeting siRNAs from nanoparticles caused a reduction in synovial inflammation in an arthritic animal model, since the Notch pathway is related with the release of pro-inflammatory cytokines [7]. Nevertheless, the use of these molecules is still limited for clinical applications since some parameters such as long-term safety and the specific guidance to the target cell or tissue remain unclear [8]. In this sense, the release of controlled doses of low-molecular-weight drugs such as statins, glucocorticoids and some antibiotics is preferred due to their increased stability [9]. For example, the incorporation of the antibiotic azithromycin, which has been described to have anti-inflammatory properties, into calcium-phosphate-coated polycaprolactone membranes lead to a decrease in the number of the M1 type macrophages, as well as an increase in the M2 type macrophages [10]. However, it has been previously described that the release of low-molecular-weight drugs occurs faster compared to molecules with higher molecular weights, such as proteins [11]. This could lead to an increase in the circulating concentrations of the drug in vivo, with a higher risk of exceeding the toxicity threshold and causing side effects [12].

Alternatively, ions have been described as plausible candidates compared to other molecules due to their therapeutic potential and stability. Moreover, multiple ion-releasing biomaterials have been developed, achieving a wide range of release profiles and released concentrations [13]. However, there is limited evidence showing the anti-inflammatory potential of divalent ions. A previous study showed that supplementing cell culture medium with Cu^2+^ concentrations below 10 µM could have an influence in macrophage polarization, promoting the expression of markers associated to M2 phenotype [14]. This fact was further confirmed by several studies releasing low concentrations of Cu^2+^ from biomaterials [15]. In a similar way, Mg^2+^ at concentrations of 3.2 and 12.8 mM were shown to increase M2-related markers such as CD206 and IL-10 [14,16]. Despite the promising results, in the presence of the pro-inflammatory stimulus LPS, ions showed a limited ability to induce an anti-inflammatory response [14]. A feasible strategy that could be used to improve the immunomodulatory potential of these ions and modulate macrophage response might be combining therapeutic concentrations of these ions with low, non-cytotoxic doses of anti-inflammatory drugs. In this sense, dexamethasone (dex) is a model drug that has been described as an anti-inflammatory molecule that can modulate the immune response and, more precisely, the activation of macrophages [17]. Previous studies have suggested that only dex concentrations above 100 nM are able to reduce the production of pro-inflammatory cytokines such as TNF-α and IL-1β, as well as other pro-inflammatory mediators [18,19]. Nevertheless, some reports have also shown moderate effects at concentrations as low as 0.1 nM [20]. Moreover, recent studies have suggested that Cu^2+^ and Mg^2+^ are able to modulate macrophage response acting on the NFκB and PI3K/Akt pathways, respectively [21,22], while dex modulates the MAPK, NFκB or STAT pathways after binding to the glucocorticoid receptor (GR), suggesting that these molecules could modulate macrophage response through different intracellular actions [21,22]. Thus, the combination of these ions with dex could lead to a combined M2-stimulating effect.

The aim of this study was to analyze the effect of bioactive ions Cu^2+^ and Mg^2+^ in combination with low dex concentrations on macrophage polarization, to determine a possible synergistic improvement in the anti-inflammatory potential of these molecules. To further analyze the possible synergistic effect, both ions were tested in the absence and in the presence of a pro-inflammatory environment to further test the anti-inflammatory potential of these dex and ion combinations.

## 2. Materials and Methods

### 2.1. Cell Culture and Differentiation

THP-1 monocytic cell line was obtained from DSMZ (ACC 16; Braunschweig, Germany) and cultured in RPMI 1640 medium (Sigma; St. Louis, MO, USA) supplemented with 10% FBS (Sigma; St. Louis, MO, USA) and 1% penicillin–streptomycin (Fisher Scientific; Waltham, MA, USA) at a cellular density of 3 × 10^5^ cells/mL. THP-1 cell differentiation into macrophages was performed by seeding 3 × 10^4^ cells/cm^2^ and exposing them to 10 ng/mL phorbol 12-myristate 13-acetate (PMA, Sigma; St. Louis, MO, USA) for 8 h, as previously described [23]. Once 95–100% of macrophages adhered to the well plate, they were washed with DPBS and incubated with cell media that contained 10 µM Cu^2+^ (copper(II) chloride dihydrate, Sigma; St. Louis, MO, USA) and 3.2 mM Mg^2+^ (magnesium chloride, Sigma; St. Louis, MO, USA) alone or in combination with 0.1 nM dex (Sigma; St. Louis, MO, USA) for 24 and 48 h at 37 °C, 5% CO_2_ and 95% humidified air. The concentrations of Cu^2+^ and Mg^2+^ were defined as moderately anti-inflammatory in a previous study [14]. The conditions where the cells were stimulated with Cu^2+^ or Mg^2+^ supplemented medium were designated as Cu and Mg, respectively, while the conditions including the ions as well as Dex were named CuD and MgD. As a positive control of the pro-inflammatory stimulus, cells were stimulated with 100 ng/mL LPS (Sigma; St. Louis, MO, USA) to induce an acute response, while cells cultured in the absence of ions were used as a negative control of inflammation (TCP). Cells stimulated with 0.1 nM dex without ions were also used as control (Dex).

### 2.2. Macrophage Metabolic Activity Assay

Metabolic activity was assessed by performing a resazurin reduction assay. Briefly, cells seeded in a 48 well plate were incubated with 200 µL of cell culture medium containing 10 µg/mL resazurin sodium salt (Sigma; St. Louis, MO, USA) for 3 h at 37 °C, 5% CO_2_. Then, 100 µL were transferred to a 96-well plate and absorbance was measured at 570 and 600 nm using a microplate reader (Synergy HT Multi-detection Microplate Reader, Bio-Tek; Winooski, VT, USA). Cell metabolic activity was expressed in terms of percentage reduction of resazurin and then normalized to control values obtained from the TCP.

### 2.3. Macrophage Morphology Analysis

Some studies have suggested that macrophage morphology could be an indicator of the phenotype they adopt, with M1 macrophages adopting a round shape and M2 macrophages an elongated shape [24]. Therefore, the cell morphology was observed and the number of elongated cells was quantified. The morphology of differentiated THP-1 macrophages was analyzed by optical microscopy using an Olympus CKX41 microscope with a Nikon DS-Fi1 camera (Tokyo, Japan). To perform morphology measurements, 5 different regions per condition with at least 100 cells per region were analyzed using ImageJ 1.52p software. Cell morphology was determined by analyzing the aspect ratio of each cell, which is the ratio between the length of the major axis and the length of the minor axis of a cell. As previously reported, aspect ratios higher than 2.5 were associated with elongated cells [25]. The number of elongated cells and the total amount of cells was quantified in each condition and the % of elongated cells was compared to the TCP.

### 2.4. Macrophage Gene Expression Analysis

Gene expression of specific macrophage markers was evaluated by quantitative PCR (RT-qPCR). Total RNA from differentiated THP-1 cells was isolated after 24 and 48 h of cell culture with Nucleospin^®^ RNA/protein kit (Macherey-Nagel, Allentown, PA, USA) following the manufacturer’s instructions. Quantification and quality assessment of extracted RNA was performed using a microvolume plate (Take3, Bio-Tek; Winooski, VT, USA) to measure the absorbance ratio of wavelengths 260/280 nm in a microplate reader (Synergy HT Multi-detection Microplate Reader, Bio-Tek; Winooski, VT, USA). Ratios ≈2 were considered pure RNA. Reverse transcription of the RNA into cDNA was performed using Transcriptor cDNA Synthesis Kit (Roche; Basilea, Switzerland) and following the manufacturer’s protocol for cDNA synthesis with anchored oligo(dT)18 primers and random hexamer primers using a T100 Thermal Cycler (BioRad; Hercules, CA, USA). To detect target mRNAs, quantitative PCR was performed with QuantiNova SYBR Green PCR Kit (Qiagen; Hilden, Germany) following the manufacturer’s protocol. Briefly, 10 ng of cDNA per reaction were amplified under the following conditions: an initial activation step of 2 min at 95 °C, denaturation for 5 s at 95 °C and 40 cycles of annealing/extension for 10 s at 60 °C in a BioRad CFX96 Real-Time System. The primer sequences of the target genes that were analyzed are detailed in Table 1. Gene expression was normalized using the mean threshold cycle (Ct) value of the housekeeping gene β-actin. The 2^−ΔΔCt^ method was used to compare the mRNA expression levels between different groups. The results of fold expression were compared to the TCP in each timepoint. Then, the results of fold expression were transformed in log2 data and the mean values were represented in heatmaps using Morpheus matrix visualization and analysis software. Moreover, the M1/M2 ratio was calculated from the expression of CCR7 and CD206 markers as described elsewhere [26]. This ratio can be used to determine the overall proportion of cells of each phenotype. Values of the M1/M2 ratio greater than 1 indicate that the predominant phenotype is M1, while values lower than 1 indicate that the predominant phenotype is M2.

### 2.5. Mitigating the Effect of a Pro-Inflammatory Stimulus with Ions and Dexamethasone

The changes in macrophage polarization induced by dex and its combination with copper and magnesium ions were analyzed in the presence of a pro-inflammatory stimulus to identify a possible synergistic effect in the reduction of the inflammatory response. Macrophages were differentiated as previously described and simultaneously exposed for 24 and 48 h to the pro-inflammatory molecule LPS alone or in combination with either dex, copper and magnesium or copper and magnesium in combination with dex (as described in Table 2). The conditions where the cells were simultaneously stimulated with LPS and Dex, Cu^2+^ or Mg^2+^ supplemented medium were designated as DexL, CuL and MgL, respectively. The conditions including the ions, Dex and LPS were named as CuDL and MgDL. The concentration of LPS used in this experiments was 10 ng/mL to generate a mild pro-inflammatory response as reported in previous studies [14]. Macrophage response was analyzed by metabolic activity and gene expression following the protocols that have been previously described.

### 2.6. Statistical Analysis

A metabolic activity assay was carried out in quadruplicate, while gene expression analysis was performed in triplicate. Data are expressed as mean ± standard deviation. The results were analyzed using a non-parametric test as normal distribution and equality of variances were not confirmed. Kruskal–Wallis for multiple comparison analysis and Mann–Whitney for one-to-one comparison were performed. The statistical analysis was performed using MINITAB^®^ (version 18, Minitab Inc., State College, PA, USA). Statistical significance was accepted at *p* < 0.05.

## 3. Results

### 3.1. Macrophage Metabolic Activity Assay

First, a resazurin reduction assay was performed to analyze the metabolic activity of the cells cultured with Cu^2+^ and Mg^2+^ ions, as well as with the combination of dex with Cu^2+^ (CuD) and Mg^2+^ (MgD).

As observed in Figure 1, none of the conditions induced significant reductions (*p* < 0.05) in metabolic activity compared to the TCP, which had values around 100%, regardless of the combination of ions and dex. However, some of the conditions were able to induce a significant increase in the metabolic activity compared to dex. At 24 h, only MgD was able to induce a significant increase in the metabolic activity of the cells (*p* < 0.05), whereas at 48 h Cu, Mg and MgD conditions induced this increase.

### 3.2. Macrophage Morphology Analysis

The effect of Cu^2+^, Mg^2+^, dex and the combination of dex with both ions on macrophage morphology was analyzed. The majority of macrophages presented a round shape with some elongated cells, regardless of the time in culture and the stimuli used in each condition (Figure 2a). The proportion of elongated cells was then quantified in each condition. No significant changes were observed in the proportion of elongated cells when comparing with the TCP or with dex (Figure 2b).

### 3.3. Macrophage Gene Expression Analysis

To observe a possible synergistic effect in macrophage polarization caused by the combination of the anti-inflammatory drug dex with Cu^2+^ and Mg^2+^, the gene expression of M1 and M2 markers was determined (Figure 3 and Appendix A). The expression of each gene was compared to the TCP (with a log2 value of 0) to observe the variation in these markers.

First, the expression of M1- and M2-related markers was analyzed in the conditions containing Cu^2+^ ions (Figure 3a). As expected, the positive control for inflammation (LPS) stimulated the expression of the three pro-inflammatory M1 markers that were analyzed. Comparing the expression of M1 markers TNF-α and IL-1β with an unstimulated control (TCP), at 24 h, a significant reduction (*p* < 0.05) was observed in the Dex, Cu and CuD conditions. At 48 h, only the CuD condition maintained a lower expression of TNF-α, while Cu induced a significant increase in its expression (*p* < 0.05). IL-1β expression was still reduced in the conditions containing dex. Regarding the expression of the M1 marker CCR7, at 24 h, only CuD was able to induce a significant decrease (*p* < 0.05). When macrophages were stimulated for 48 h with dex alone, as well as Cu, an increase in CCR7 was observed, while CuD showed a decrease in its expression. Comparing the expression of M1-related markers with the dex control, at 24 h, Cu induced a significantly higher expression of TNF-α and IL-1β. However, at 48 h, Cu induced a significantly higher expression of TNF-α (*p* < 0.05) while, interestingly, no significant differences were found in IL-1β expression. Moreover, at 48 h, CuD was able to maintain a reduced expression of TNF-α, IL-1β and CCR7.

Furthermore, the expression of anti-inflammatory M2 markers was analyzed and compared to the TCP. On the one hand, at 24 and 48 h, Cu and CuD induced a significant increase in IL-10 expression. Dex, however, only induced an increase in IL-10 at 48 h. On the other hand, CuD and dex were able to upregulate the expression of TGF-β both at 24 and 48 h, while Cu only induced and increase in this marker at 48 h. Moreover, both Dex and CuD, at 24 and 48 h, induced a great increase in CD206 expression. However, Cu only induced a significant increase of this marker at 48 h. Comparing the expression of these markers with Dex alone, at 24 h, Cu induced a significant increase in the expression of IL-10 (*p* < 0.05) while no significant differences were observed at 48 h. CuD increased the expression of IL-10 and TGF-β at 24 h, while at 48 h a significant decrease was observed in both markers. Lastly, at 24 and 48 h both Cu and CuD induced a significantly lower expression (*p* < 0.05) of the marker CD206 compared to dex.

The expression of M1- and M2-related markers was then analyzed in those conditions containing Mg^2+^ ions (Figure 3b). Comparing the expression of M1-related markers to the TCP, at 24 h, MgD was able to induce a significant decrease in the expression of TNF-α (*p* < 0.05) and, at 48 h, it caused a decrease in IL-1β expression. Mg did not induce any significant changes in the expression of these markers at either of both time points. However, at 24 and 48 h, Mg was able to induce a significant increase in the expression of CCR7 (*p* < 0.05), while at 48 h, MgD induced a decrease in this marker. Comparing the expression of these markers with dex, no differences were found in TNF-α. However, both at 24 and 48 h, Mg induced a higher expression of IL-1β. Moreover, at 24 h, Mg was able to induce a significantly higher expression of CCR7, while at 48 h, MgD significantly reduced the expression of this marker.

Regarding the expression of M2-related markers compared to the TCP, at 24 and 48 h, Mg and MgD promoted the expression of IL-10 and CD206. At 48 h, these conditions also promoted the expression of TGF-β. Comparing the expression of these markers to dex, it could be observed that at 24 h Mg and MgD increased IL-10 expression. Moreover, at 48 h, MgD induced a lower expression of TGF-β, and Mg showed a reduced expression of CD206 at 24 and 48 h. MgD did not induce any significant differences regarding the expression of CD206 compared to dex.

As a first approach to analyzing macrophage phenotype, the M1/M2 ratio was calculated from the expression of CCR7 and CD206 (Figure 4). As expected, dex, CuD and MgD were able to significantly reduce the M1/M2 ratio below 1, while the positive control for inflammation (LPS) greatly increased the M1/M2 ratio above 1. Cu and Mg conditions induced values of the M1/M2 ratio that were not significantly different from the TCP (*p* > 0.05), although they were significantly higher compared to dex. Nevertheless, this ratio did not allow for the identification of significant differences between dex and CuD and MgD conditions.

### 3.4. Mitigating the Effect of a Pro-inflammatory Stimulus with Ions and Dexamethasone

To further test the anti-inflammatory potential of the combination of dexamethasone with copper and magnesium ions, THP-1 macrophages were cultured in the presence of 10 ng/mL of LPS combined with either dex alone or dex combined with ions. First, the cellular metabolic activity was analyzed with a resazurin reduction assay (Figure 5). An increase in metabolic activity was observed at 24 h when cells were cultured with dex and MgDL and the results were compared to the TCP and LPS. Moreover, MgDL also induced a significant increase compared to dex both at 24 and 48 h. Then, cell morphology was observed by optical microscopy (Appendix A). The majority of the cells presented a round morphology, with few cells adopting a spindle-like shape. No apparent differences were observed in cell shape when comparing between conditions.

Next, the expression of M1 markers was analyzed and compared to the response caused by LPS alone (Figure 6a,b and Appendix A). As expected, LPS was able to induce the expression of TNF-α, IL-1β and CCR7. Moreover, DexL was able to induce a significant decrease in the expression of TNF-α at both 24 and 48 h (*p* < 0.05), as well as a reduction in IL-1β and CCR7 at 48 h compared to LPS. Then, the expression of M1 markers in LPS-activated macrophages by Cu^2+^ ions alone or in combination with dex (CuL and CuDL) was analyzed. At 24 h, CuL significantly decreased the expression of TNF-α and CCR7 markers, while no significant differences were found at 48 h. Interestingly, CuDL did cause a significant increase in the expression of TNF-α at 48 h (*p* < 0.05). At 24 and 48 h, CuL and CuDL downregulated the expression of IL-1β. When the gene expression of M1-related markers was compared to dex, both CuL and CuDL induced a higher expression of TNF-α at 24 and 48 h. At 24 h, CuL and CuDL induced a significantly lower expression of IL-1β (*p* < 0.05) and, at 48 h, they induced an increase in CCR7 expression.

Furthermore, the expression of M2-related markers IL-10, TGF-β and CD206 was also assessed. Comparing the expression of these markers with LPS, at 24 h, DexL was able to increase the expression of IL-10 and TGF-β. Moreover, at 24 and 48 h, DexL was also able to strongly upregulate the expression of CD206. These results were expected due to the anti-inflammatory potential of this drug. When the LPS-activated macrophages were cultured in the presence of copper ions, at 24 h, CuDL was able to induce an increase in IL-10 expression, although there was a significant decrease in this marker at 48 h (*p* < 0.05). CuDL also induced a significant decrease in TGF-β at 48 h (*p* < 0.05). Moreover, CuDL increased the expression of CD206 at 24 and 48 h, while CuL only increased the expression of this marker at 24 h. When these results were compared to DexL, CuDL induced a significant decrease in IL-10 and CD206 expression at 24 h, and downregulated the expression of TGF-β at 24 and 48 h.

Then, the gene expression of LPS-activated macrophages in the presence of Mg^2+^ (MgL) or when combiningMg^2+^ with dex (MgDL) was analyzed (Figure 6b). Comparing the expression of M1 markers to the response caused by LPS alone at 24 and 48 h, MgDL induced a significant decrease in TNF-α and IL-1β expression (*p* < 0.05), while MgL did not induce any significant changes in the expression of these markers. However, MgL did induce an increase in CCR7 expression at 24 h, followed by a decrease at 48 h. Comparing the expression of the M1 markers to dex, MgDL induced a decrease in IL-1β at 24 h and a decrease in TNF-α at 48 h. Furthermore, at 48 h, both Mg and MgDL increased the expression of CCR7.

Regarding the expression of M2 markers, at 24 h, MgL and MgDL significantly increased IL-10 and CD206 expression (*p* < 0.05). At 48 h, MgDL decreased IL-10 expression, while maintaining the increased expression of CD206. Moreover, no differences were found in TGF-β expression compared to LPS in any of these conditions. Comparing these results with DexL, at 24 h, MgDL downregulated IL-10 and CD206 expression, while at 48 h no significant differences were observed. In addition, at 24 and 48 h, MgL induced a lower expression of CD206.

When the M1/M2 ratio was analyzed, the value significantly decreased below 1 when macrophages were cultured with dex and with the combination of dex and the ions (Figure 7). The values of the ratio in these conditions were significantly lower compared to the LPS (*p* < 0.05). Moreover, at 24 h, CuL induced a decrease in this ratio compared to LPS, while at 48 h, this decrease was observed in MgL. Both CuL and MgL promoted higher values of this ratio compared to dex alone. However, no significant differences in the ratio were observed when macrophages cultured with the combination of dex and the ions (CuDL and MgDL) were compared to dex alone.

## 4. Discussion

Among the different strategies to fabricate biomaterials that induce tissue regeneration, molecule-eluting scaffolds are an interesting approach. Among the different molecules or cues eluted, divalent ions have gained interest in recent years, since they are stable cues that have already shown some benefits when released from scaffolds. Nevertheless, few reports have investigated the association between the release of these ions and macrophage response. As already shown in our previous report, 10 µM Cu^2+^ and 3.2 mM Mg^2+^ were able to induce an M2-like phenotype [14]. However, in the presence of a mild pro-inflammatory stimulus such as 10 ng/mL LPS, these ions were unable to significantly reduce the expression of M1 markers, indicating a limited potential of Cu^2+^ and Mg^2+^ to control an ongoing pro-inflammatory response. In this study, we wanted to analyze the effect of Cu^2+^ and Mg^2+^ therapeutic ions on macrophage polarization and supplement its effect with other anti-inflammatory cues, such as low doses of the anti-inflammatory drug dexamethasone to improve their anti-inflammatory potential.

Dex is a synthetic glucocorticoid with the ability to inhibit TNF-α and IL-1β cytokine production in LPS-stimulated macrophages [27]. TNF-α inhibition takes place as a consequence of the suppression of the activity of tumor necrosis factor converting enzyme (TACE) via the p38α MAPK pathway, while IL-1β reduction is caused by an inhibition in the nuclear translocation of transcription factors NFκB/Rel and AP-1 [18,19]. Since Cu^2+^ and Mg^2+^ have been described to have an effect in NFκB transcription factor through the activation of a MyD88 independent pathway and PI3K/Akt pathway, respectively [21,22], we investigated whether the combination of these ions with dex could lead to a synergistic effect that could stimulate the expression of M2 markers, while inhibiting pro-inflammatory marker expression.

Our findings showed that the combination of dex with the ions did not significantly reduce the metabolic activity of macrophages, indicating that the concentrations used were not cytotoxic (Figure 1). Moreover, several studies have associated a round shape of macrophages with the M1 phenotype, while a spindle-like shape was associated with an M2 phenotype [24,28]. However, when we performed the morphology analysis, we could observe that macrophage shape did not change when cells were cultured with LPS (M1 stimulus) or dex (M2 stimulus), showing that cell elongation is not an indicator of macrophage polarization (Figure 2). However, previous reports have shown that different chemical stimuli such as macrophage colony-stimulating factor (M-CSF) and granulocyte-macrophage colony stimulating factor (GM-CSF), which are used to induce the maturation of macrophages from peripheral blood, can influence the cellular shape [29,30]. Indeed, this could indicate that the presence of elongated cells could be dependent on the type of stimuli they receive rather than the phenotype they adopt.

Regarding gene expression, the M1/M2 ratio was analyzed as a first approach. This ratio greatly decreased below 1 when macrophages were cultured with dex and the combination of dex and the ions (Figure 4), indicating that dex was able to promote M2 polarization. When the expression of each gene was observed (Figure 3), macrophages stimulated with dex showed a decrease in TNF-α and IL-1β markers at 24 h, as previously reported [31]. Moreover, an overall increase in IL-10, TGF-β and CD206 expression was observed when macrophages were cultured with dex. These results suggest that dex alone is able to induce macrophage polarization towards an M2 phenotype. Furthermore, CuD was able to significantly reduce the expression of the M1 markers TNF-α, IL-1β and CCR7 at 48 h compared to dex, suggesting that copper ions might be improving the anti-inflammatory actions of dex on these markers. On the other hand, MgD decreased CCR7 and stimulated IL-10 expression compared to dex, suggesting that the combination of 3.2 mM of Mg^2+^ with dex had some effect in potentiating the anti-inflammatory effect of dex.

To further investigate the anti-inflammatory potential of the combination of ions with dex, macrophages were simultaneously stimulated with 10 ng/mL of LPS and with dex alone or combined with Cu^2+^ and Mg^2+^ (Figure 6). As previously described by other authors, dex was able to significantly reduce TNF-α and IL-1β while increasing IL-10, TGF-β and CD206 expression compared to macrophages stimulated with LPS [19,32]. The M1/M2 ratio further confirmed the M2-polarizing effect of dex in a pro-inflammatory environment (Figure 7). However, CuDL induced an increase in TNF-α expression at 48 h, as well as a decrease in IL-10 and CD206 at 24 h and a significant decrease in TGF-β expression at 48 h compared to dex. These results suggest that the presence of Cu^2+^ might be interfering with the anti-inflammatory action of dex through a mechanism that should be further studied. Nevertheless, CuDL also induced a strong reduction in the expression of IL-1β at 24 h, suggesting that copper ions could be initially blocking the expression of this marker. Regarding the effect of MgDL, a significant decrease in IL-1β, IL-10, TGF-β and CD206 expression compared to dex was observed at 24 h; however, no significant differences were observed when macrophages were stimulated for 48 h, meaning that the presence of magnesium might be delaying the effect of dex on gene expression of both pro-inflammatory and anti-inflammatory markers.

The changes observed in the expression of M1- and M2-related genes in the presence of dex and in the absence and in the presence of the ions could be related to the activation of the pathways associated with M1 or M2 polarization. The activation of the M1 or M2 phenotype is highly dependent on the type of stimulus, since each type of stimulus can be detected by a receptor that will activate the corresponding signaling pathways [33,34]. On the one hand, M1 macrophage polarization in response to an infectious challenge with LPS occurs through the activation of the Toll-like receptor 4 (TLR4), which is involved in the activation of transcription factors NFκB (p50 and p65 heterodimer) and the interferon regulatory factor 5 (IRF5) among others [35]. In turn, the NFκB pathway is involved in the transcription of pro-inflammatory cytokines such as TNF-α and IL-1β [36,37]. A schematic illustration of this pathway can be found in Appendix A.

M2 macrophage polarization occurs in response to the activation of other receptors such as IL-10R and IL-4Rα, as well as the glucocorticoid receptor (GR). The main pathways involved in this process are PI3K-Akt and JAK-STAT, which activate STAT3, STAT6 and mTORC1 transcription factors and induce anti-inflammatory cytokine expression [38,39]. In the case of macrophages stimulated with glucocorticoids such as dex, the activation of the GR leads to both genomic and non-genomic effects. The GR is able to enter the nucleus and to modify gene expression by directly binding to the DNA, associating with STAT3 to act as a coactivator or binding to the NFκB heterodimer to act as a corepressor. Moreover, the GR also induces the activation of the PI3K-Akt pathway in the cytoplasm to elicit a rapid cellular response [40]. A recent study showed that the GR is able to bind NFκB response elements in the DNA to inhibit the pro-inflammatory response [41].

Little has been published regarding the effect of magnesium supplementation on the intracellular signaling pathways of macrophages. The anti-inflammatory effect of Mg^2+^ could be related to the activation of the TRPM7-PI3K-Akt axis in macrophages. In this pathway, the cation channel TRPM7 allowed the entrance of Mg^2+^, which induced the activation of PI3K and the phosphorylation and activation of Akt-1, even in the presence of LPS. Moreover, Mg^2+^ downregulates the TLR4-NFκB pathway [21,42]. This effect of Mg^2+^ in NFκB was confirmed when macrophages were stimulated with extracts of fibrinogen and magnesium [43]. Interestingly, Akt1 protein is able to bind Mg^2+^ in its active site, which has a great influence on the catalytic activity of this protein [44]. As already discussed, the combination of MgD had some effect on the expression of the inflammatory markers in comparison to dex. This could be related to the fact that both Mg^2+^ and dex activate the PI3K-Akt pathway, thus inducing a similar type of response in macrophages in the absence of another stimulus. However, in LPS-activated macrophages, a delay in the anti-inflammatory response was observed compared to dex, indicating that Mg^2+^ could delay either the inhibition of NFκB by the GR or the catalytic activity of Akt1 protein. Nevertheless, further studies analyzing these pathways should be performed to confirm the interaction of Mg^2+^ in the presence of dex.

Additionally, little evidence has been found regarding the possible mechanisms responsible for the anti-inflammatory potential of low concentrations of Cu^2+^. Previous studies have shown that Cu^2+^ is able to induce the activation of NFκB [22,45]. However, macrophages were stimulated with high concentrations of Cu^2+^, which promote macrophage polarization to the M1 phenotype. So far, there is no evidence regarding the activation of pathways by lower concentrations of Cu^2+^, which are able to induce an M2 phenotype as suggested by recent studies [15,46,47]. Intracellular Cu^2+^ homeostasis is regulated by several proteins such as CTR1 (copper transporter), ATP7 and the COMMD family of proteins. Moreover, COMMD proteins act as regulators of NFκB, promoting its ubiquitination and degradation, and regulate the activity of ATP7 transporter, which is implicated in regulating the cytosolic concentration of Cu^2+^ [48,49]. It has been described that Cu^2+^ overload for long times in liver cells can cause a decrease in COMMD protein expression, allowing for the expression of pro-inflammatory genes associated with this pathway [50]. Moreover, a recent study observed that releasing Cu^2+^ from Ti6Al4V implants caused an increase in COMMD1 expression, which was associated with a decrease in NFκB expression [51]. Therefore, different Cu^2+^ concentrations could modulate NFκB response through the activity of COMMD proteins. As discussed before, the combination of Cu^2+^ and dex decreased M1 marker expression, suggesting a possible interaction in GR binding to glucocorticoid response elements in the DNA. Interestingly, the combination of Cu^2+^ with dex in LPS-activated macrophages induced an increased TNF-α response, indicating a possible lack of inhibition of NFκB by the GR as a consequence of Cu^2+^. However, more experiments should be performed to check NFκB activation and GR receptor binding in these conditions.

## 5. Conclusions

Overall, we observed that the combination of Cu^2+^ ions and dex was able to effectively reduce the expression of pro-inflammatory markers compared to dex. However, the synergistic anti-inflammatory effect of the combination of Mg^2+^ and dex could not be confirmed as there were no major differences in comparison to dex alone. Moreover, combining Cu^2+^ ions and dex in the presence of a pro-inflammatory stimulus reduced the expression of IL-1β, while at later time points it induced a strong TNF-α response compared to dex. The combination of Mg^2+^ ions and dex in the presence of a pro-inflammatory stimulus also did not promote any improvement in comparison to dex alone. This study provides initial evidence of a positive effect on macrophage response of the combination of divalent ions and low concentrations of anti-inflammatory drugs with some limitation under a pro-inflammatory environment. The results obtained in this study could be relevant for tissue engineering applications, more precisely for the design of novel biomaterials with a dual release of divalent ions and small molecules.

## Figures and Tables

**Figure 1 biomedicines-10-00764-f001:**
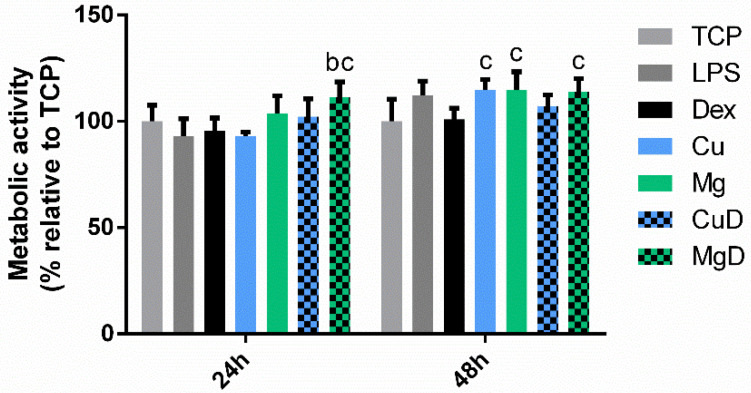
Effect of dex and the combination of dex with Cu^2+^ and Mg^2+^ on THP-1 cell metabolic activity. THP-1 cells were cultured with Cu^2+^ and Mg^2+^ ions (Cu and Mg, respectively), as well as with a combination of dex with Cu^2+^ (CuD) and Mg^2+^ (MgD) for 24 and 48 h. Metabolic activity results were represented as percentage relative to an unstimulated control (TCP) and compared with the TCP of each day. Statistically significant differences compared to TCP, LPS and dex were indicated with b and c, respectively (*p* < 0.05).

**Figure 2 biomedicines-10-00764-f002:**
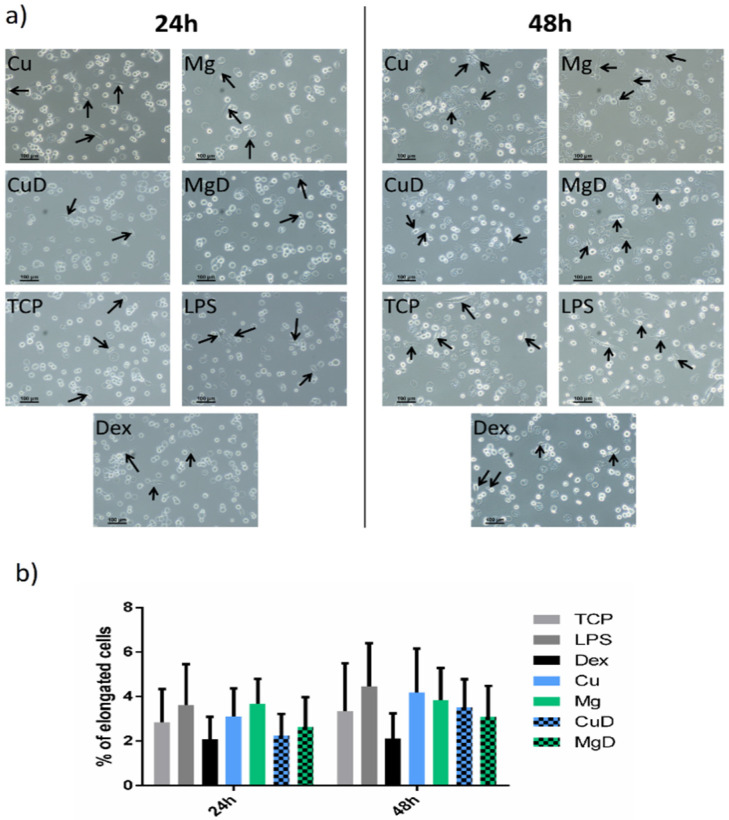
Effect of dex and the combination of dex with Cu^2+^ and Mg^2+^ on THP-1 cell morphology. (**a**) Elongated cells, which were defined as those in which the aspect ratio was higher than 2.5, are indicated with black arrows. Scale bars = 100 µm. (**b**) Quantitative analysis of THP-1 cell morphology. The number of elongated cells (aspect ratio higher than 2.5) in each condition was represented.

**Figure 3 biomedicines-10-00764-f003:**
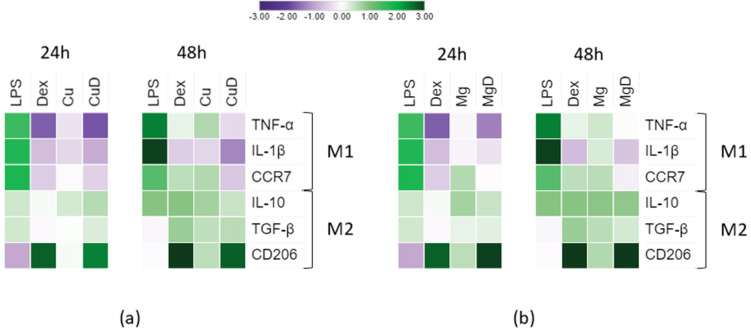
Effect of dex and the combination of dex with Cu^2+^ and Mg^2+^ on the expression of M1 and M2 markers in the THP-1 macrophage cell line. (**a**) Gene expression after incubating macrophages with Cu^2+^ (Cu) and the combination of Cu^2+^ with dex (CuD). (**b**) Gene expression after incubating macrophages with Mg^2+^ (Mg) and the combination of Mg^2+^ with dex (MgD). After 24 or 48 h of incubation, total RNA was extracted and the mRNA levels of three M1 markers (TNF-α, IL-1β and CCR7) and three M2 markers (IL-10, TGF-β and CD206) were determined by RT-qPCR. Results were normalized against β-actin mRNA and are expressed relative to the mRNA levels of unstimulated THP-1 macrophages (TCP). The complete statistical analysis is in Appendix A.

**Figure 4 biomedicines-10-00764-f004:**
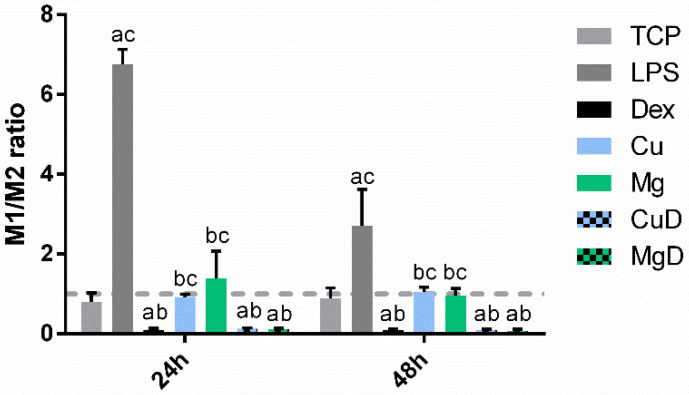
M1/M2 ratio of macrophages stimulated with dex and the combination of dex with Cu^2+^ and Mg^2+^. M1/M2 ratio was calculated from the gene expression of CCR7 and CD206 markers. The results of the M1/M2 ratio for each condition were compared to the TCP of each day. Statistically significant differences compared to TCP, LPS and dex were indicated with a, b or c, respectively (*p* < 0.05).

**Figure 5 biomedicines-10-00764-f005:**
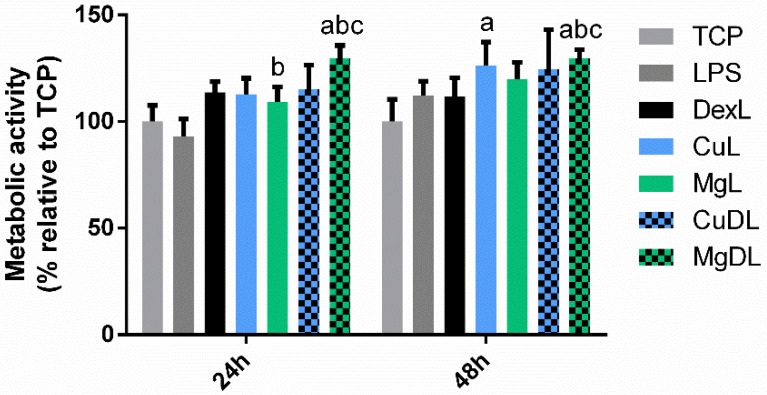
Effect of dex and the combination of dex with Cu^2+^ (CuDL) and Mg^2+^ (MgDL) on LPS-activated THP-1 cell metabolic activity. THP-1 cells were cultured with different dex and a combination of dex with Cu^2+^ and Mg^2+^ (CuDL and MgDL, respectively) for 24 and 48 h. Metabolic activity results were represented as percentage relative to an unstimulated control (TCP) and compared with the TCP of each day. Statistically significant differences compared to TCP, LPS (10 ng/mL) and dex were indicated with a, b and c, respectively (*p* < 0.05).

**Figure 6 biomedicines-10-00764-f006:**
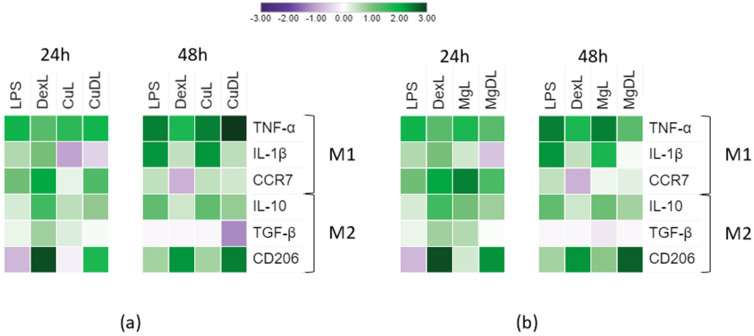
Effect of dex and the combination of dex with Cu^2+^ and Mg^2+^ on the expression of M1 and M2 in LPS-activated THP-1 macrophage cell line. (**a**) Gene expression after incubating LPS-activated macrophages with Cu^2+^ (CuL) and the combination of Cu^2+^ with dex (CuDL). (**b**) Gene expression after incubating LPS-activated macrophages with Mg^2+^ (MgL) and the combination of Mg^2+^ with dex (MgDL). After 24 or 48 h of incubation, total RNA was extracted and the mRNA levels of three M1 markers (TNF-α, IL-1β and CCR7) and three M2 markers (IL-10, TGF-β and CD206) were determined by RT-qPCR. Results were normalized against β-actin mRNA and are expressed relative to the mRNA levels of unstimulated THP-1 macrophages (TCP). The complete statistical analysis is in Appendix A.

**Figure 7 biomedicines-10-00764-f007:**
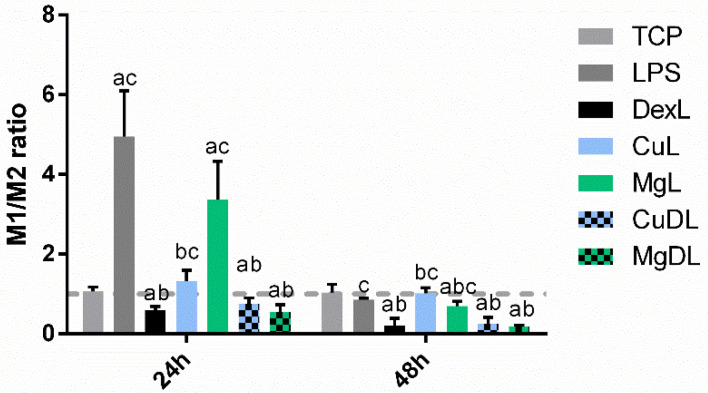
M1/M2 ratio of LPS-activated macrophages stimulated with dex and the combination of dex with Cu^2+^ and Mg^2+^. Statistically significant differences compared to TCP, LPS and dex were indicated with a, b or c, respectively (*p* < 0.05).

**Table 1 biomedicines-10-00764-t001:** Primer sequences for qRT-PCR.

Macrophage Phenotype	Gene	Forward (Sequence 5′–3′)	Reverse (Sequence 5′–3′)
M1	TNF-α	TTCCAGACTTCCTTGAGACACG	AAACATGTCTGAGCCAAGGC
	IL-1β	GACACATGGGATAACGAGGC	ACGCAGGACAGGTACAGATT
	CCR7	GGCTGGTCGTGTTGACCTAT	ACGTAGCGGTCAATGCTGAT
M2	IL-10	AAGCCTGACCACGCTTTCTA	ATGAAGTGGTTGGGGAATGA
	TGF-β	TTGATGTCACCGGAGTTGTG	TGATGTCCACTTGCAGTGTG
	CD206	CCTGGAAAAAGCTGTGTGTCAC	AGTGGTGTTGCCCTTTTTGC
Housekeeping	β-actin	AGAGCTACGAGCTGCCTGAC	AGCACTGTGTTGGCGTACAG

**Table 2 biomedicines-10-00764-t002:** Summary of the molecules incorporated into the cell culture medium in the different conditions that were analyzed.

	Molecules Incorporated Into the Cell Culture Medium			
	LPS (10 ng/mL)	Dex (0.1 nM)	Cu^2+^ Ions (10 µM)	Mg^2+^ Ions (3.2 mM)
TCP	−	−	−	−
LPS	+	−	−	−
DexL	+	+	−	−
CuL	+	−	+	−
MgL	+	−	−	+
CuDL	+	+	+	−
MgDL	+	+	−	+

## Data Availability

The datasets generated and analyzed in the current study are available from the corresponding author upon reasonable request.

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
