# Peer review of "Modulation of Macrophage Response by Copper and Magnesium Ions in Combination with Low Concentrations of Dexamethasone"

_biomedicines, 2022, doi:10.3390/biomedicines10040764_

Round 1

Reviewer 1 Report

In this study, Díez-Tercero et al. investigated whether low concentrations of dexamethasone (dex) and its combination with cationic ions (Cu, Mg) could synergistically modulate the macrophage phenotype. The study is interesting, but the manuscript is difficult to follow. Moreover, the paper is not written in international standard methods. For instance, abbreviations should be first described in full (e.g., line 163: qRT-PCR). Another example is that the authors use dex to describe dexamethasone, but later in the text, they again write dexamethasone (line 256). There is no consistency or standardization in their writing. Additionally, in Materials and Methods, following the description of a material or equipment, the name of the company as well as the city and country of its headquarter should be clearly described. Basically, all of them are missing. Moreover, “ml” should be written as “mL”. There must have a space between numbers and their meaning: e.g., “100μm” should be written as “100 μm”; “24h” should be “24 h”; etc.. A detailed and professional revision of English grammar and typographic errors should be done.

Other points that need revision:

The abstract does not introduce the entire concept and results of the study. Please write a well-structured abstract. Please check the instruction for authors for details.

It is unclear how the authors defined the concentrations of Cu and Mg ions.

In Figures 4 and 7, the negative control is missing. This is a critical error.

Line 119: “The aim of this project”. Please replace “project” for “study”.

Figure 2 and 3 should be combined. There is no meaning to be separated. Figure 3, are there any significant difference among the groups?

It is unclear why the authors use Kruskal–Wallis and Mann–Whitney tests. Since qRT-PCR, or metabolic activity analyses give numerical values, they should use one-way or two-way ANOVA. Kruskal–Wallis and Mann–Whitney tests are definitely not appropriate.

Line 144: “for 3h at 37C, 5% CO2 Then” . A dot is missing between CO2 and Then.

Discussion, lines 429-430: …10 μM Cu2+ and 3.2 mM Mg2+ were able to induce an M2-like phenotype. The results of this study do not match with those of previous studies. In Figures 3 and 8, there are no difference in M1/M2 ratio with Cu or Mg stimulation.

Conclusions: The authors’ conclusions are not supported by the data. They need to rephrase it.

Author Response

Dear Reviewer,

We thank the reviewer for the comments and the suggestions. We have carefully gone through each comment made and a point-by-point response to your comments are below:

In this study, Díez-Tercero et al. investigated whether low concentrations of dexamethasone (dex) and its combination with cationic ions (Cu, Mg) could synergistically modulate the macrophage phenotype. The study is interesting, but the manuscript is difficult to follow.

Moreover, the paper is not written in international standard methods. For instance, abbreviations should be first described in full (e.g., line 163: qRT-PCR). Another example is that the authors use dex to describe dexamethasone, but later in the text, they again write dexamethasone (line 256). There is no consistency or standardization in their writing.

Author’s response: We thank the reviewer for addressing this issue. We have revised the manuscript and verified that all of the abbreviations are correctly described.

Additionally, in Materials and Methods, following the description of a material or equipment, the name of the company as well as the city and country of its headquarter should be clearly described. Basically, all of them are missing.

Author’s response: We have added the name of the company, as well as the city for each material described in Materials and Methods.

Moreover, “ml” should be written as “mL”. There must have a space between numbers and their meaning: e.g., “100μm” should be written as “100 μm”; “24h” should be “24 h”; etc.

Author’s response: We have carefully checked and corrected these typographic errors.

A detailed and professional revision of English grammar and typographic errors should be done.

Author’s response: An English native speaker revised the manuscript to correct the mistakes regarding the English grammar.

Other points that need revision:

The abstract does not introduce the entire concept and results of the study. Please write a well-structured abstract. Please check the instruction for authors for details.

Author’s response: we agree with the reviewer that the abstract did not correctly capture the results of this study, we have modified it.

It is unclear how the authors defined the concentrations of Cu and Mg ions.

Author’s response: we agree with the reviewer that the rationale behind using these concentrations was unclear. These concentrations of Cu and Mg ions are based on a previous study published by our group, where we could observe that these conditions were able to induce the M2 macrophage phenotype (https://doi.org/10.1038/s41598-021-91070-0). We have provided a reference to this study in Materials and Methods.

In Figures 4 and 7, the negative control is missing. This is a critical error.

Author’s response: these two figures show the results of gene expression quantified by RT-qPCR. The negative control used in both cases was the TCP (unstimulated THP-1 cells), which was used to compare the expression of each gene at each timepoint. Therefore, the value of the log2 transformed data for the TCP is 0 and would be represented in the figure as blank squares. Nevertheless, we have specified this fact in the Results section.

Line 119: “The aim of this project”. Please replace “project” for “study”.

Author’s response: we thank the reviewer for the suggestion, we have replaced the word “project”.

Figure 2 and 3 should be combined. There is no meaning to be separated. Figure 3, are there any significant difference among the groups?

Author’s response: we thank the reviewer for the suggestion, we have combined both figures. Since no significant differences were observed among the different groups, we have eliminated the following sentence from the figure caption: “Statistical significance was accepted at p<0.05”

It is unclear why the authors use Kruskal–Wallis and Mann–Whitney tests. Since qRT-PCR, or metabolic activity analyses give numerical values, they should use one-way or two-way ANOVA. Kruskal–Wallis and Mann–Whitney tests are definitely not appropriate.

Author’s response: we performed a normality analysis, where we could observe that the distribution of the data did not follow a normal distribution. We also performed an equal variances test, which indicated that not all of the variances were equal. Therefore, parametric tests such as ANOVA could not be used and we chose to perform non-parametric statistics.

Line 144: “for 3h at 37C, 5% CO2 Then”. A dot is missing between CO2 and Then.

Author’s response: we have added the missing dot

Discussion, lines 429-430: …10 μM Cu2+ and 3.2 mM Mg2+ were able to induce an M2-like phenotype. The results of this study do not match with those of previous studies. In Figures 3 and 8, there are no difference in M1/M2 ratio with Cu or Mg stimulation.

Author’s response: this statement is based on the evaluation of the gene expression profile shown in Figures 3 and 6, as well as the M1/M2 ratio in Figure 7. In Figure 3, for example, both Cu and Mg were able to stimulate the expression of all of the M2 related markers, while Cu induced a moderate reduction in the expression of IL-1β. Moreover, in Figure 7, CuL and MgL conditions induced a significant reduction of the ratio compared to LPS alone, therefore indicating that the ions are exerting a moderate M2 polarizing and anti-inflammatory effect.

Conclusions: The authors’ conclusions are not supported by the data. They need to rephrase it.

Author’s response: we have rephrased the conclusions to match the results reported in this study.

Reviewer 2 Report

The authors presented a detailed study to understand the effects of dexamethasone in combination with copper and magnesium ions.

Author Response

Dear Reviewer,

Thank for your revision, we checked some minor spells and grammatical errors.

Reviewer 3 Report

The major flaw in the study presented in the submitted manuscript relates to the motivation behind the use of dexamethasone in bone formation. Although dexamethasone has been the magic bullet in the induction of osteogenic differentiation in vitro, its in vivo use has been implicated osteoporosis. Actually the orthopedics literature clearly implicates the exposure of individuals to dexamethasone that is often prescribed to moderate inflammation in osteoporosis. Actually osteoporosis patients medical history links the condition in 20% of osteoporosis cases directly to the exposure in glucocorticoids. Interestingly, researchers speculating that the osteoporotic event is linked in limited differentiation of MSC to osteoblasts in the presence of dexamethasone lead to the discovery in 1988 of the recipe (that included 10-100nM dex) the induced strong osteoblastic differentiation in rat and later human MSC in vitro. This surprising finding corresponds only to in vitro cultures while in vivo in a systemic environment still glucocorticoids are blamed for osteoporosis. I have seen many researchers as in this case confusing the in vitro success of dex inducing osteoblastic differentiation with a potential controlled release approach in vivo that can be beneficial in bone healing. Clearly the authors are correct at approaching the dex use with suppression of inflammation linked to macrophages but the links with bone formation mentioned throught the intro and discussion are misleading. 

The study itself is fairly well conducted and the findings, although based on very moderate effects, support the beneficial use of the combined use of dex with Cu2+ ions moderates the expression of proinflammatory markers. As the goal was to implement the strategy in a controlled release methodology applicable in implantation it was expected to see a similar approach that could provide initial indicators on the release profile. At a minimum level, variable levels of dex should have been used to assess the influence of the dual use of dex with Cu2+, having a culture of differentiation MSC being used to assess ostoblastic differentiation on top. Here we see only macrophages, one level of dex (and that level is unclear as the authors mention 0.1 nM and 0.1 ng/ml that are obviously VERY DIFFERENT concentrations). In the intro tha authors also mention that lower than 100nM dex are needed for osteoblastic differentiation of MSC a fact that is true for rat MSC where 10nM are adequate but incorrect for human MSC that necessitate 100 nM for robust osteoblastic differentiation.

Overall, the introduction and scope are very misleading, the bone references are not reflecting all aspects of the issue, and the study is driven by incorrect motives. It is also conducted using only one level of dex and in the absence of a control release strategy and a MSC culture differentiating (for the same concentrations of the additives) to osteoblasts. I am not sure of the experimental part with the macrophages can be salvaged after rewriting the intro and posing a new motivation that relates to inflammatory drugs use alone.

Author Response

Dear Reviewer,

We thank the reviewer for his comments that allowed us to realize that we may not reflected well the motivation of the original manuscript. The aim of this study was to increase knowledge about the possible effect of combining divalent ions with moderate anti-inflammatory potential in combination with a model anti-inflammatory drug, dexamethasone. We do not attempt to develop a potential treatment using dexamethasone, as we were concerned about the potential issues with its clinical translation. The current study attempts to explore synergistic effect between copper and magnesium ions with a model drug that is widely used: 10.1089/ten.tea.2020.0287, 10.1039/D0BM01142H, 10.1242/dmm.037887, 10.1016/j.biomaterials.2016.11.004…

Therefore, we have carefully checked and updated the manuscript to clarify the main motivation of this study according to the valuable information provided by the reviewer, in order to focus on the inflammatory response induced by copper and magnesium ions, as well as low concentrations of dex. We have also corrected some issues, such as the concentration of dex used in this study, which was unclear in the text and in Table 2.

Moreover, the reviewer suggested that a controlled release methodology could have been described in our study. In this sense, we thank the reviewer for this valuable suggestion. This manuscript tries to provide some knowledge prior to generate a dual drug delivery system that releases ions and small molecules. In fact, we have already developed this platform and assessed its biological effect on both the inflammatory and osteogenic response, which will lead to another publication.